

# Spatial variability in soil pH and land use as the main influential factor in the red beds of the Nanxiong Basin, China

Ping Yan[1], Hua Peng[1,†], Luobin Yan[2], Shaoyun Zhang[1], Aimin Chen[1] and Kairong Lin[1]

[1] School of Geography and Planning, Sun Yat-sen University, Guangzhou, Guangdong Province, China
[2] School of Geographical Sciences, Southwest University, Chongqing, Chongqing Province, China
[†] Deceased.

## ABSTRACT

Soil pH is the main factor affecting soil nutrient availability and chemical substances in soil. It is of great significance to study the spatial variability of soil pH for the management of soil nutrients and the prediction of soil pollution. In order to explore the causes of spatial variability in soil pH in red-bed areas, the Nanxiong Basin in south China was selected as an example, and soil pH was measured in the topsoil by nested sampling (0–20 cm depth). The spatial variability characteristics of soil pH were analyzed by geostatistics and classical statistical methods, and the main factors influencing spatial variability in soil pH are discussed. The coefficient of variation in the red-bed areas of Nanxiong Basin was 17.18%, indicating moderate variability. Geostatistical analysis showed that the spherical model is the optimal theoretical model for explaining variability in soil pH, which is influenced by both structural and random factors. Analysis of the spatial distribution and pattern showed that soil pH is relatively high in the northeast and southwest, and is lower in the northwest. These results indicate that land use patterns and topographic factors are the main and secondary influencing factors, respectively.

## INTRODUCTION

Soil pH is an indicator of the acidity or alkalinity of soil, and is a reflection of important physical and chemical properties determining soil quality (*Nagy & Kónya, 2007*). Soil pH also has a profound impact on a number of other soil properties. Extremes in acidity or alkalinity will change the nutrients available and result in the unbalanced absorption of elements in plants (*Zhao et al., 2011*).

Spatial heterogeneity refers to the lack of homogeneity and the complexity in the distribution in space of the properties of a system (*Nagy & Kónya, 2007*). The spatial heterogeneity of soil parameters such as pH and content of organic matter and of nitrogen, phosphorus and potassium, has an important influence on the distribution and spatial pattern of plants (*Stoyan et al., 2000*; *Augustine & Frank, 2001*; *Silvia et al., 2016*). Studying

Corresponding author
Luobin Yan, yanluobin@swu.edu.cn

the spatial heterogeneity and the driving factors behind soil properties is significant for revealing ecosystem function and biodiversity (*Augustine & Frank, 2001*).

With the continuous development of geographic information technology, studying the spatial variability of soil properties using a combination of geostatistics and GIS (Geographical Information System) technology has become one of the most important topics in the different fields in which soil is investigated (*Romano, 1993*; *Foroughifar et al., 2013*). In conventional soil survey soil properties are recorded at representative sites and assigned to entire mapping unit, which are delineated using both physio-graphic and geopedologic approaches (*Shit, Bhunia & Maiti, 2016*). Although soil surveyors are very well aware of the spatial variability of soil properties, conventionally prepared soil maps do not reflect it as soil units are limited by boundaries (*Heuvelink & Webster, 2001*). In addition, the conventional method of soil analysis and interpretation are laborious, time consuming, hence becoming expensive. Starting near the end of the 1970s, scholars worldwide applied geostatistics to study the spatial variability of soil properties (*Trangmar, Yost & Uehara, 1986*).

Geostatistics is a widely used method for studying the spatial distribution of regionalized variables (*Liu, Shao & Wang, 2012*; *Emadi et al., 2016*; *Mohamed et al., 2018*). Many scholars have studied the spatial distribution characteristics of various soil properties by this method (*Zhang & Li, 2002*; *Zhang & Li, 2010*; *Liu, Shao & Wang, 2011*; *Turgut & Öztaş, 2012*; *Liu, Shao & Wang, 2013a*). However, most of these studies were limited to a single terrain (*Huang et al., 2012*; *Zhao et al., 2017*), vegetation type *Riha, Senesac & Pallant, 1986*; *Zaremehrjardi, Taghizadehmehrjardi & Akbarzadeh, 2010*), land use (*Mao et al., 2014*; *Miheretu & Yimer, 2017*) or other environmental factors, but analysis of them simultaneously is still lack.

Previous research has revealed that spatial variation in soil pH controls off-season $N_2O$ emission in agricultural soils (*Russenes et al., 2016*), however, soil properties vary in space and time across natural ecosystems (*Bogunovic et al., 2017a*; *Bogunovic et al., 2017b*; *Griffiths et al., 2017*), and distributions of soil nutrients and related environmental factors depend on scale. Many studies have shown that soil pH is negatively correlated with many variables, such as soil organic carbon content, total nitrogen content, total phosphorus content, precipitation, temperature, and clay content (*Liu, Shao & Wang, 2013b*). Because the spatial distribution of soil pH has structural and stochastic characteristics, measuring it accurately has implications for crop production (*Liu, Shao & Wang, 2013b*). *Reijonen, Metzler & Hartikainen (2016)* demonstrated that soil pH dictates the accessibility of vanadium V(+V) and V(+IV), by investigating the chemical bioavailability of vanadium species. Therefore, it is important to study the spatial variability of soil pH on a regional scale together with the factors influencing it; these are important for the regulation of soil acidity and alkalinity, control of environmental pollution, and sustainable utilization and management of soil nutrients in addition to the ensemble of components of the regional ecological environment.

In China, the soil that forms on red beds is known as 'purple soil' (*Yan et al., 2017*). According to the results of the 34-province-wide soil census, the total area of purple soil is $2.17 \times 10^5 \, km^2$ (*Atsumoto et al., 2015*). Many studies have shown that the purple soil formed

on red-bed parent material is the most seriously eroded of all soil types in the Yangtze River Basin (*Wang et al., 2009*; *Li et al., 2008*). This is especially visible in humid regions,where severe erosion can threaten the sustainable development of agriculture in South China (*Yan et al., 2017*) The change in soil structure and the removal of topsoil resulting from erosion may cause the loss of nutrients and environmental degradation, thereby inhibiting plant growth (*Sheoran, Sheoran & Poonia, 2010*). The change in availability of soil nutrients affects not only crop production and vegetation growth, but also the structure of the ecological environment (*Jin & Jiang, 2002*; *Zhang et al., 2010*). The factors affecting the spatial variability in soil pH in red-bed areas have not received much study. Therefore, studying the spatial distribution characteristics of soil pH is important for the sustainable utilization and management of soil nutrients and to improve soil productivity.

This study was carried out in a red-bed area in China with the following objectives: (i) to assess the value of the soil pH; (ii) to reveal the spatial variability and the environmental influencing factors.

## MATERIALS AND METHODS

### Study area

Nanxiong Basin (24°35′–25°24′N, 113°50′–114°44′E) is a narrow basin located in the northeast of Guangdong Province, China (Fig. 1). The elevation ranges from 48 to 1,421 m above sea level (ASL). The subtropical monsoon climate here is characterised by long hot summers and short winters. The average temperature is 19.6 °C and the annual precipitation and potential evaporation are 1,555.1 mm and 1,678.7 mm, respectively (*Yan et al., 2017*). The total area of Nanxiong Basin is 3,692 km$^2$. The rainy season is from March to August. The dense river networks in Nanxiong Basin belong to the Pearl River systems. Purple soil accompanies the red-bed parent material distributed in the central part of the basin. Nanxiong Basin is a red-bed basin with a severe soil erosion problem due to its dominant purple-soil texture (Calcaric Regosols in the FAO taxonomy); the red beds occupy an area of 1,500 km$^2$ andare mainly distributed in the central part of the basin. Land use mainly consists of farmland, shrubland, woodland, and grassland. The main vegetation communities are mixed forest of *Pinus massoniana* Lamb and broadleaf trees, secondary forest with mixed deciduous and broadleaf trees, and mainly artificial *Eucalyptus* and pine forests (Fig. 2, *Yan et al., 2017*). According to the 2009 1:5,000 land-use map from the Shaoguan Municipal Bureau of Land and Resources, land-use types (Fig. 3) mainly include woodland in the Southeast part, farmland in stretched west to east in central part, bareland in the north and southwest part.

### Research method
#### Soil sample collection
Samples were collected in November 2017 after crops (mainly rice) were fully harvested. A total of 225 samples were gathered from 0–20 cm depth by the nested sampling method at sampling densities. Soil samples were air-dried and passed through a 2 mm sieve for laboratory analysis of soil pH was measured in a 1:2.5 soil:water (DI water) suspension using a PP-50-P11 pH meter (with measurement error ± 0.002) (*Liu, Shao & Wang, 2013c*)

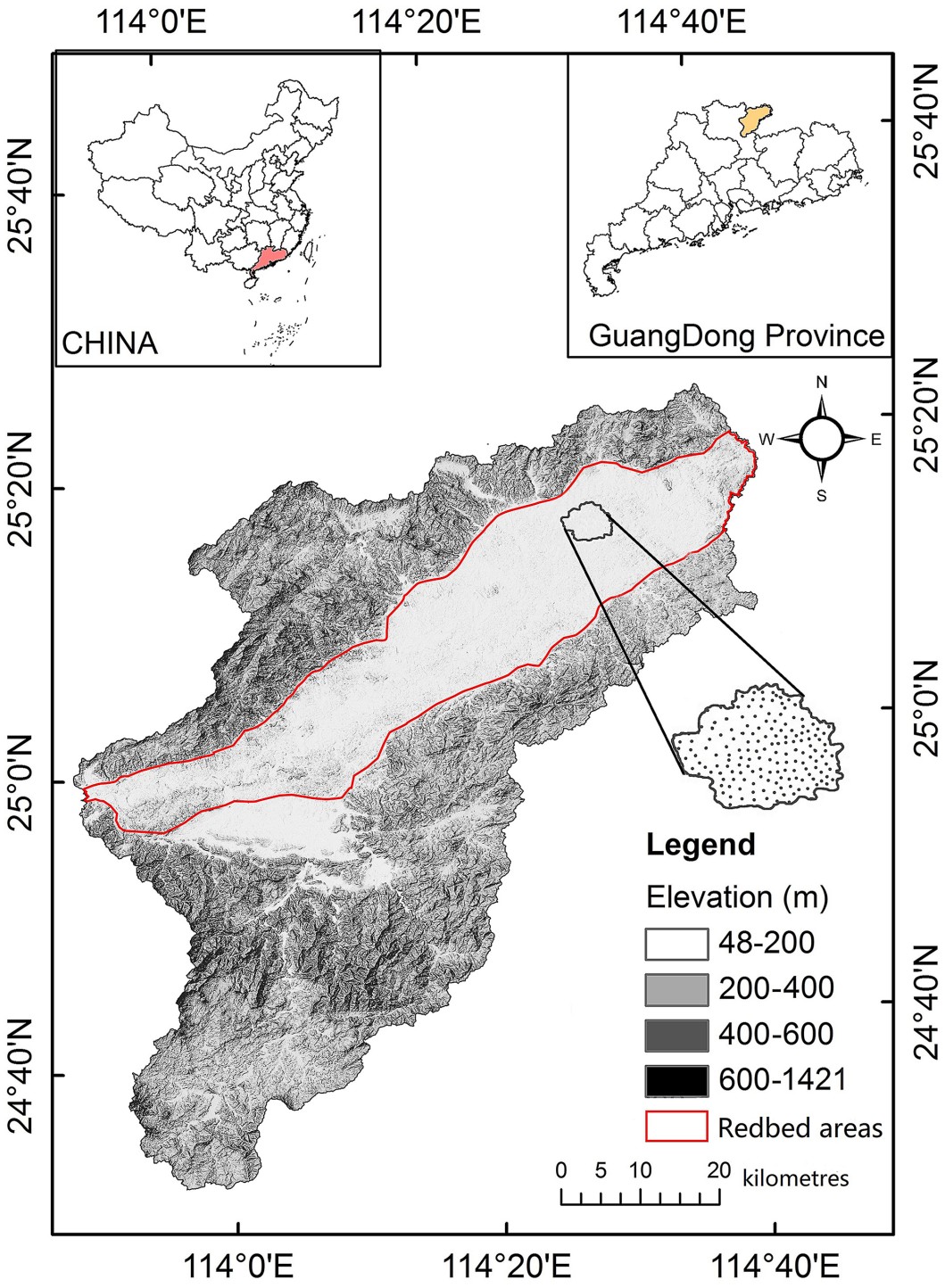

**Figure 1  Location map of the study area.** Adapted from *Yan et al. (2017)*. Reprinted with permission: this is an open access article distributed under the Creative Commons Attribution License (CC BY 4.0).

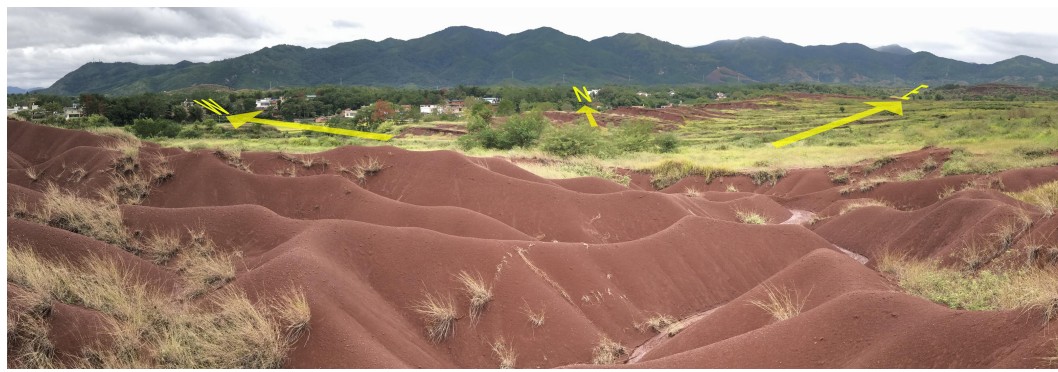

**Figure 2** **Location map of sampling point.** Photo by Ping Yan.

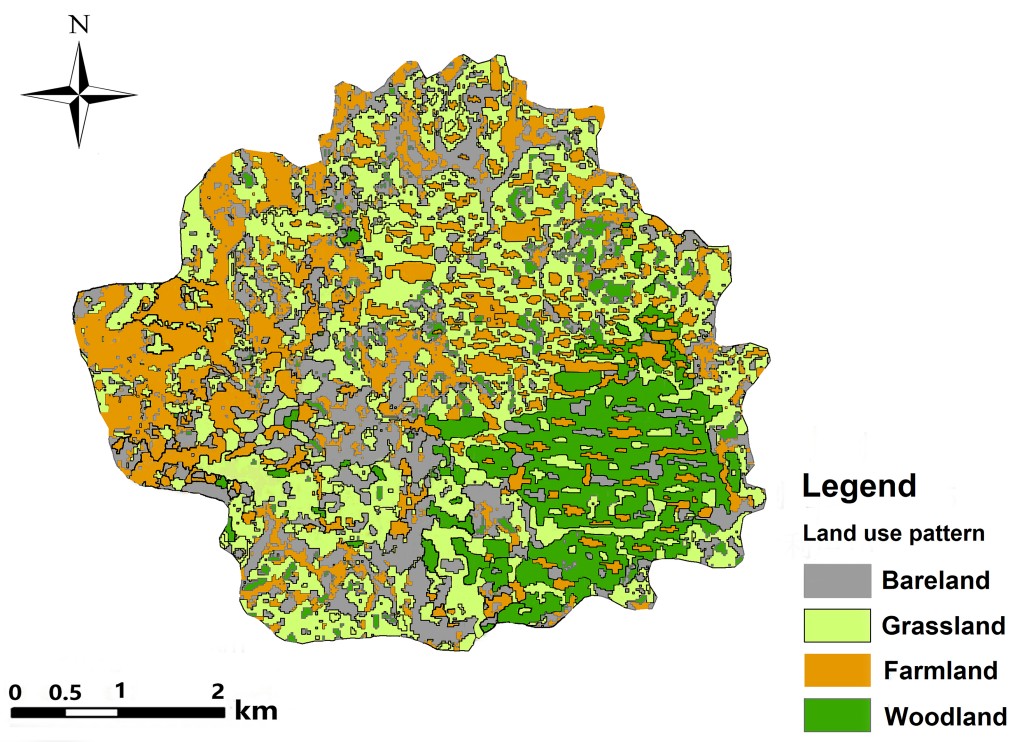

**Legend**

**Land use pattern**

- Bareland
- Grassland
- Farmland
- Woodland

**Figure 3** **Land-use map of Nanxiong Basin in 2010.**

equipped with a calibrated combined glass electrode (*Bogunovic et al., 2017a*; *Bogunovic et al., 2017b*). The global positioning system (GPS) (5-m precision) was then used to identify the site's longitude, latitude and elevation. The actual sampling sites were chosen to represent the main topography, land use, and vegetation types within the range of vision. Site slope and aspect measured with a geological compass, and information on human activities (irrigation, fertilizer use and crop yield) was collected from surveys of the local inhabitants. The distribution of sample points is shown in Fig. 1.

*Data analysis*

Some basic statistics were calculated, such as the minimum, maximum, and mean values of measurements and their coefficient of variation (CV). The Kolmogorov–Smirnov (K–S) test and correlation analysis of soil pH with topographical variables were performed to analyse data distribution, using the statistical software SPSS 19 (IBM SPSS Statistics for Windows, IBM Corp., Armonk, NY, USA). GS+7 (Gamma Design Software, Plainwell, MI, USA) was used to do the geostatistical analysis. The K–S method was used to evaluate data normality and asymmetry in terms of skewness and kurtosis because these factors have important implications for the performance of the interpolation methods.

A semivariogram is the basic tool of geostatistics (*Oliver & Webster, 1986*; *Goovaerts, 1999*; *Nasseh et al., 2016*). The formula used to calculate the semivariogram is:

$$\gamma(h) = \frac{1}{2N(h)} \sum_{i=1}^{N(h)} [Z(xi) - Z(xi+h)]^2. \tag{1}$$

In Eq. (1), N(h) is the logarithm of the distance when the distance equals $h$, and $Z(x_i)$ is the value at location $x_i$; $Z(x_i + h)$ is the value at a distance $h$ from $x_i$ (*Yang et al., 2016*; *Rosemary et al., 2017*). Appropriate model functions were fitted to the semivariograms. The semivariograms were used to determine the degree of spatial variability on the basis of the classes of spatial dependence distinguished by *Cambardella et al. (1994)*, strong spatial dependence ($C_0/(C_0 + C) > 75\%$), moderate spatial dependence ($25\% < C_0/(C_0 + C) < 75\%$) and weak spatial dependence ($C_0/(C_0 + C) < 25\%$). In ArcGIS 9.2 (ESRI 2006. ArcGIS Desktop: Redlands, CA: Environmental Systems Research Institute.), we used kriging interpolation in the geostatistics module to draw the spatial distribution map of soil pH and the trend analysis chart in order to analyse the characteristics of the spatial variability. The main factors controlling spatial variation in soil pH and their influence were analysed using maps of the soil type, slope, aspect, elevation, and land use.

## RESULTS

### Descriptive statistics of soil pH

Descriptive statistics of soil pH is presented in Table 1. Soil pH of the study area ranged between 7.50 and 8.50, with an average value of 8.04 and a median of 8.05. The mean soil pH for the red-bed region, which was calculated from 225 soil samples, is higher than the estimated mean soil pH for the whole of China (6.8) and lower than the mean soil pH for the Loess Plateau region (8.49). The main factors determining soil pH were the region's humid climate and the relatively high calcium carbonate content in the soft rock underlying the red beds. The criteria proposed by *Wilding (1985)* were used to classify the parameters into most (CV > 35%), moderate (CV 15–35%) and least (CV < 15%) variable classes. The standard deviation in soil pH values was 1.38 and the CV value for the pH in this area was 17.18%. Accordingly, the pH in this area could be classified as moderately variable. In general, pH is considered to be a stable soil parameter. Similar CV values were reported by *Tsui, Chen & Hsieh (2004)* ,*Fu, Tunney & Zhang (2010)*, and *Liu, Shao & Wang (2013c)* , in all these studies, variability was found to be moderate. According to the observed trend in

**Table 1  Statistical characteristic values of soil pH.**

| Soil properties | Sample size | Range | Median | Mean | Standard deviation | Skewness | Kurtosis | Coefficient of variation (%) | K-S test |
|---|---|---|---|---|---|---|---|---|---|
| pH | 225 | 7.50–8.50 | 8.05 | 8.04 | 1.38 | −0.25 | −0.42 | 17.18 | 0.10 |

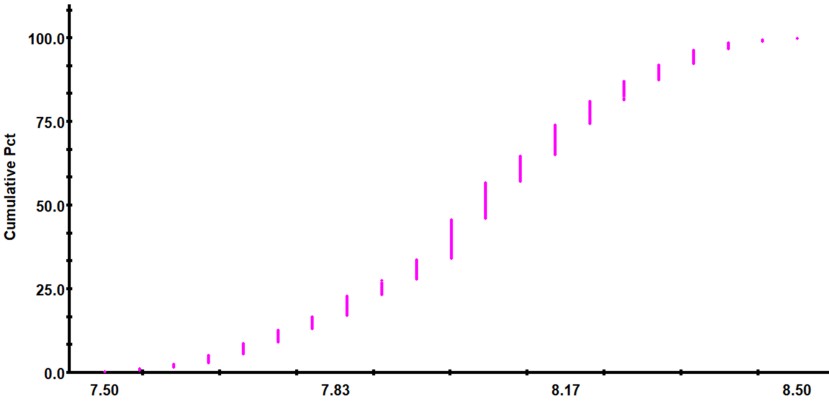

**Figure 4  Trend of the cumulative frequency of soil pH.**

the accumulation frequency of soil pH (Fig. 4), the pH value in the study area was mainly in the range of 7.95–8.20. The K–S test showed that the pH values of the sample points were normally distributed, and thus meet the requirements of geostatistics analysis (Table 1).

## Spatial variability in soil pH
### Isotropic semivariogram of soil pH

GS+7.0 software was used to fit the soil pH in the study area to the theoretical model (Table 2). The variogram's fitting model was selected based on the nugget effect, the coefficient of determination ($R^2$) and the range of variation (*Bogunovic et al., 2017a*; *Bogunovic et al., 2017b*). As can be seen from Table 2, the value for nugget ($C_0$) is 0.12, the value for sill ($C_0 + C$) is 0.18, the ratio of nugget ($C_0$) and sill ($C_0 + C$) is 66.67%, and the determination coefficient ($R^2$) is 0.812. High coefficients of determination indicate that the models fitted the semivariogram well (*Jeloudar et al., 2014*). The nugget–sill ratio of 66.67%, indicating that the soil pH had a moderate spatial dependence (*Cambardella et al., 1994*). The spherical model gave the best fit for the variation in soil pH in the study area. The main structural factors were climate, parent material and terrain; these can enhance the spatial dependency of soil pH. In contrast, random factors, which are the result of human activity such as farming and fertilization, can make the spatial dependency of soil pH weaker (*Isaaks & Srivastava, 1989*). This moderate spatial dependence of soil pH in the red beds implies that the spatial variation of soil pH in the study area is mainly caused by both structural and random factors.

As can be seen in Fig. 5, when the separation distance is more than 161 m, the semivariance fluctuates only slightly, and then stabilizes. This trend might be caused by differences in directional variation. The variance at 250 m implies that the range of

**Table 2 Isotropic semivariogram theory model and related parameters of soil pH.**

| Soil property | Theoretical model | Nugget ($C_0$) | Sill ($C_0+C$) | Nugget/Sill (%) | Range (m) | Determining coefficient ($R^2$) |
|---|---|---|---|---|---|---|
| Soil pH | Spherical model | 0.12 | 0.18 | 66.67 | 161 | 0.812 |

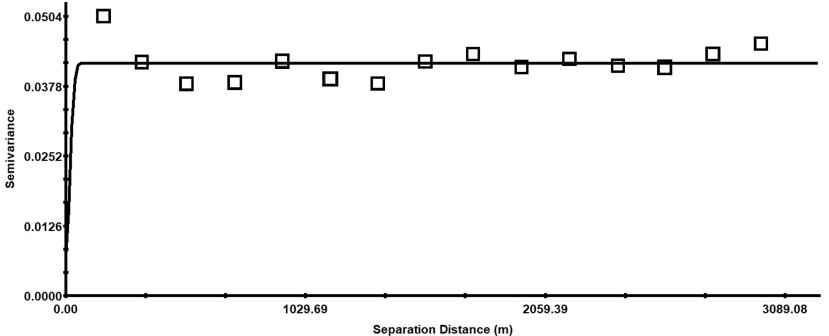

**Figure 5 Isotropic semivariance of soil pH.**

the spatial dependence is much wider than the sampling interval. Therefore, the current sampling design was appropriate for this study.

In order to understand the characteristics of the spatial variation in soil pH, the semivariogram was drawn in four directions, E–W (0°), NE–SW (45°), S–N (90°) and SE–NW (135°), using the GS+7.0 software. As shown in Fig. 6, the spatial variation exhibits large differences in different directions, showing the heterogeneity. Table 3 shows that the best-fitting models in the four directions are all spherical. The nugget ($C_0$) and sill ($C_0+C$) values are different and their ratio ranges from 60.24% to 69.23%, indicating moderate variation.

As shown in Fig. 6, the range of the soil pH values from the northeast to the southwest (45°) and from the southeast to the northwest (135°) is significantly smaller than that from east to west (0°) and from north to south (90°), indicating that the variation in the 0° and 90° directions is more complex than those at 45° and 135°.

From east to west (0°), when the separation distance is greater than 161 m, the difference in the semivariance of the soil pH begins to fluctuate, first increasing and afterward decreasing to around 0.0388. The semivariance from north to south (90°) shows the same trend, alternating between high and low, but the degree of fluctuation in the E–W (0°) direction is smaller. When the separation distance is larger than 169 m, the variation of the soil pH in the NE–SW (45°) and SE–NW (135°) directions is more stable near 0.0388, and the degree of variation is not very different. The main reason is that the area is near the badlands hills in the NE–SW and the SE–NW directions; the topography and parent materials are of great influence, and in the SE–NW direction there are more hills and larger undulations. However, in the N–S and E–W directions (0° and 90°, respectively), the soil pH shows high spatial homogeneity because the relief is low and the only land use is
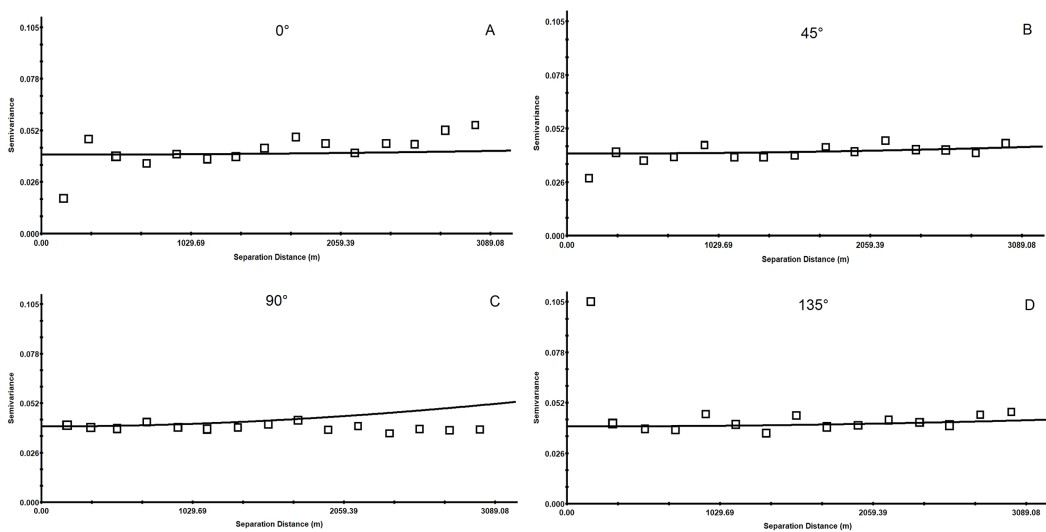

**Figure 6  Anisotropic semi-variance of soil pH.** The semivariogram of the spatial variation in soil pH was drawn in directions of E–W (0°) in (A); the semivariogram of the spatial variation in soil pH was drawn in directions of NE–SW (45°) in (B); the semivariogram of the spatial variation in soil pH was drawn in directions of S–N (90°) in (C); the semivariogram of the spatial variation in soil pH was drawn in directions of SE–NW (135°) in (D).

**Table 3  Anisotropic semivariogram theory model and related parameters of soil pH.**

| Soil property | Direction | Theoretical model | Nugget ($C_0$) | Sill ($C_0+C$) | Nugget/Sill (%) | Range (m) | Determining coefficient ($R^2$) |
|---|---|---|---|---|---|---|---|
| Soil pH | 0° | Spherical model | 0.27 | 0.39 | 69.23 | 161 | 0.539 |
| | 45° | Spherical model | 0.32 | 0.47 | 68.09 | 172 | 0.586 |
| | 90° | Spherical model | 0.29 | 0.48 | 60.42 | 169 | 0.612 |
| | 135° | Spherical model | 0.35 | 0.51 | 68.62 | 182 | 0.509 |

farmland in these directions. Taken together, the soil pH in this study area has an obvious spatial heterogeneity, which is suitable for further interpolation analysis.

### Analysis of the spatial distribution of soil pH

The effect of trends is a prerequisite for and the basis of prediction by kriging interpolation. The number of parameters that are required for kriging interpolation becomes smaller as the order of the trend effect decreases. Thus, a lower order of the trend effect can reduce error, and many scholars take the lower-order trend among two trends as the trend to be used in conducting prediction by interpolation (*Li et al., 2013*). Trend analysis can provide a study area sampling point and a three-dimensional perspective with information for the attribute value on the *z*-axis. The global trend in sampling data can be analysed from different perspectives.

As shown in Fig. 7, soil pH decreases from northeast to southwest, which is consistent with the result of semivariogram analysis. The soil pH values are higher in the northeast and southwest; this pattern can be explained by the difference in land use. In the northeastern

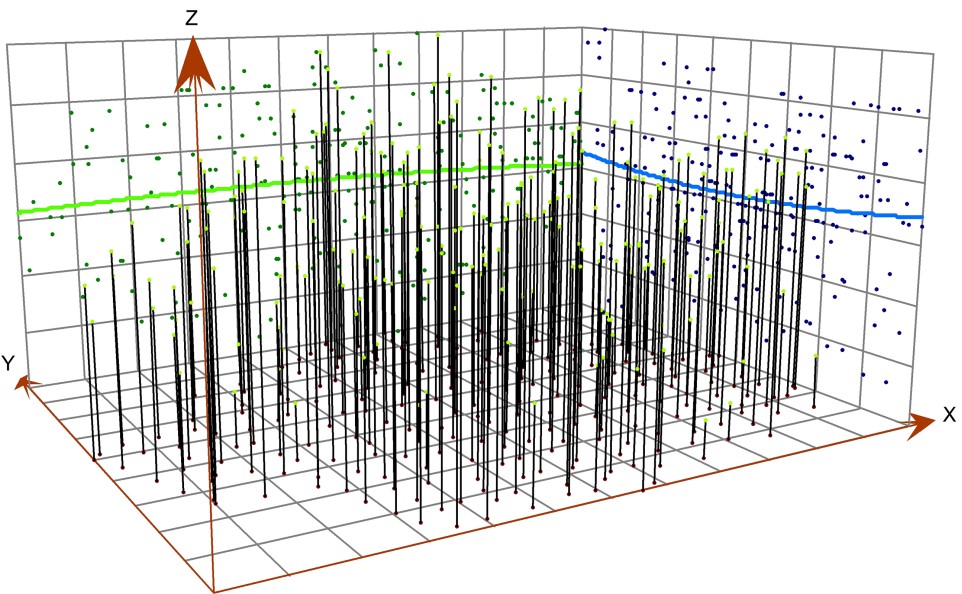

**Figure 7** **Analysis of soil pH trend.**

**Table 4** **The influence of slope and slope position on soil pH.** The difference between the letters in the same column is significant ($P < 0.05$), and the letters in brackets indicate significant difference ($P < 0.05$).

| Slope | 0–20 cm Soil layer | | |
| --- | --- | --- | --- |
| | Upper slope | Middle slope | Down slope |
| 10° | 8.41 ± 0.11a(a) | 8.39 ± 0.02a(a) | 8.01 ± 0.09b(a) |
| 15° | 8.32 ± 0.14a(a) | 8.29 ± 0.01a(a) | 8.15 ± 0.01b(a) |
| 20° | 8.09 ± 0.09b(b) | 8.02 ± 0.02b(b) | 8.26 ± 0.06ab(a) |
| 25° | 7.95 ± 0.22b(b) | 7.88 ± 0.53b(b) | 8.35 ± 0.12a(a) |

and southwestern parts, the land is unused land with a high relief. Arable land is mainly distributed in the northwest, where the relief is low and the land is strongly affected by human activities such as the use of nitrogen fertilizer, which might cause a reduction of the pH value in soil (*Yüksek et al., 2009*).

Table 4 shows that the pH value of the 0–20 cm soil layer tends to decrease from upper slope to middle slope to downslope for slope below 20°, when slope over 20°, this trend is reversed ($P < 0.05$).

### Spatial distribution pattern of soil pH

Based on the semivariance function model and the spatial distribution trend analyses, the spatial distribution pattern of soil pH in the study area was analysed by interpolation analysis of the 3D map constructed with the GS+7.0 software (*Nasseh et al., 2016*). Kriging analysis of the 3D map shows that the soil pH varies greatly in the horizontal direction in the study area (Fig. 8); the soil pH is higher in the northeast and the southwest, increases towards the southwest, and decreases towards the northwest. The result of inverse distance

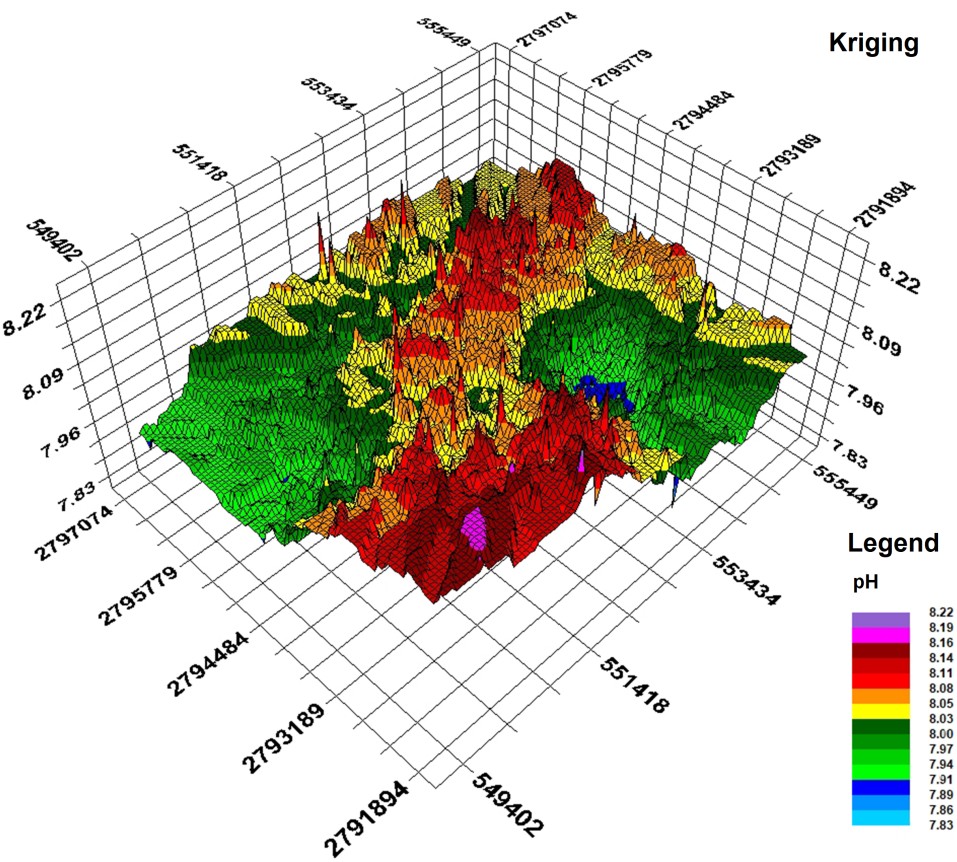

**Figure 8** Kriging interpolation map.

weighting interpolation of the 3D map shows that the overall trend for the pH in the study area is consistent with the results from kriging interpolation (Fig. 9).

## DISCUSSION

### Analysis of influential factors

Human activities and the natural environment always interact with each other. Natural factors such as climate, topography and soil properties greatly affect the way land is used by human beings and the method (*Morales et al., 2009*; *Wang, Zhang & Huang, 2009*; *Zucco et al., 2014*). In turn, the human choice for different land uses will also affect natural factors such as vegetation types and the physical, chemical and biological properties of the soil.

A large number of studies have shown that the spatial variability of soil pH is related to many factors (*Riha, Senesac & Pallant, 1986*; *Kuzel et al., 1994*; *Russenes et al., 2016*). In this study, the CV is 17.18%, which can be classified as moderate variation, and is the result of both structural factors (parent material, topography, climate) and random factors (soil biology, human disturbance, sampling design and measurement error).

Although the spatial variation of soil pH in the study area is determined by structural factors such as topographic factors and the random factors of human fertilization, it is still

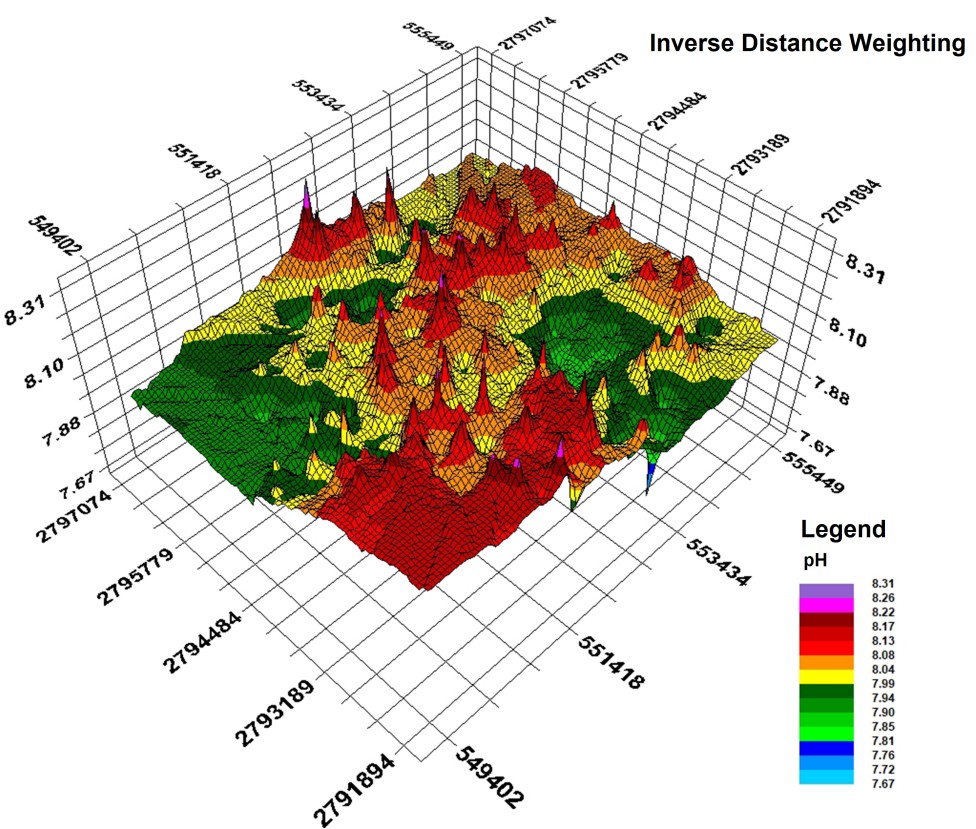

**Figure 9** Inverse distance weighting interpolation map.

not known to what extent each factor affects the spatial variation of soil pH. Therefore, two factors (topographic factors and land use) will be further discussed here to demonstrate their influence.

## Topographic factors
### Influence of slope and position along the slope on the spatial distribution of soil pH

Severe soil erosion can cause a decrease in the pH value (*Schindelbeck et al., 2008*). Due to the humid monsoon climate and the high erodibility of purple soil, which is caused by its high content of sandy particles, its pH value is generally lower than in the weathering sediments of red beds, which have a pH value higher than 8.

In general, soil pH varies significantly between different slopes and positions along the slope (*Henkel, 2003*). Thus, the pH of surface soil (0–20 cm) also varies with slope and position, reflecting the geomorphic process.

Soil properties on different slope positions were significantly affected by the degree of soil development and the leaching processes (*Tsui, Chen & Hsieh, 2004*). The effects of topographic factors on soil pH were discussed in this study. For slopes under 20° in this study, the pH of soil is the highest on the downslope, followed by that in the middle slope, and is the lowest on the upper slope. In case of ploughed vineyard decrease of
pH value along the slope catena was noticed by *Kenderessy & Lieskovsky (2014)*, which indicates that calcic carbonates could be leached from the surface layer by excess water from upper positions as a result of accelerated erosion. For slopes over 20°, soils on the footslope which is a concave position has a significantly higher pH than those on other slope positions, similar results were reported by *Tsui, Chen & Hsieh (2004)* and *Huggett (1975)*, who confirmed that slope, which is involved in the transport and accumulation of solutes, resulted in a higher pH. Thus, to some extent factors affecting soil erosion have an influence on soil pH. However, in this study case in red beds area the concrete mechanisms, how topographical factors affect the pH in different positions, needs further research.

In addition, as we know, topography is a structure factor influencing the spatial variability of soil pH. In our study area, in the E–W and N–S directions (0° and 90°, respectively), the soil pH shows high spatial homogeneity because the relief is low and the only land use is farmland. An important result is that the topography influences soil pH mainly through the slope and indirectly via effects on land use patterns; this is a general conclusion that has rarely been acknowledged elsewhere.

### Influence of aspect on the spatial distribution of soil pH

Different slope aspects experience different solar radiation, temperature and water conditions. The vegetation coverage is also different. Therefore, differences in physical, chemical, and biological processes in the topsoil are correlated with different aspect directions, leading to a heterogeneity in pH content and distribution in the topsoil (*Vieira et al., 2009*; *Salehi, Esfandiarpour & Sarshogh, 2011*). By combining the aspect distribution map of the study area and the geostatistical analysis module in the ArcGIS software, the spatial distribution map of the soil pH was analysed (Figs. 10 and 11). The result shows that the average pH value varies with aspect of the slope in the study area. The soil pH values on north- and southwest-facing slopes are relatively higher than on slopes of other aspects.

The location of the study area, in the humid red-bed area in south China, is representative for the concentrated distribution of soft rock in red beds. The best fitting models were all spherical, with a high degree of fit for the spatial variability of soil pH, and they were verified in various studies (*Wang et al., 2011*; *Liu, Shao & Wang, 2013c*), indicating that the spatial structure of soil pH in the study area was distinct.

*Kerry & Oliver (2004)* indicated that, as a rough guide, future sampling intervals should be chosen to be less than half the variogram range. According to the results of this study, future sampling intervals for monitoring pH should be 80–100 m.

### Land use pattern

Different systems of land use result in different levels of human land-use activities and have different effects on soil properties. The results showed that land use had a significant effect on surface soil pH ($P < 0.05$). As shown in Fig. 12, among the four categories of land use patterns (farmland, woodland, grassland and bare land), the average soil pH differs significantly between different land uses ($P < 0.05$). Among them, there is not much difference between woodland and grassland, though. The soil pH between different land

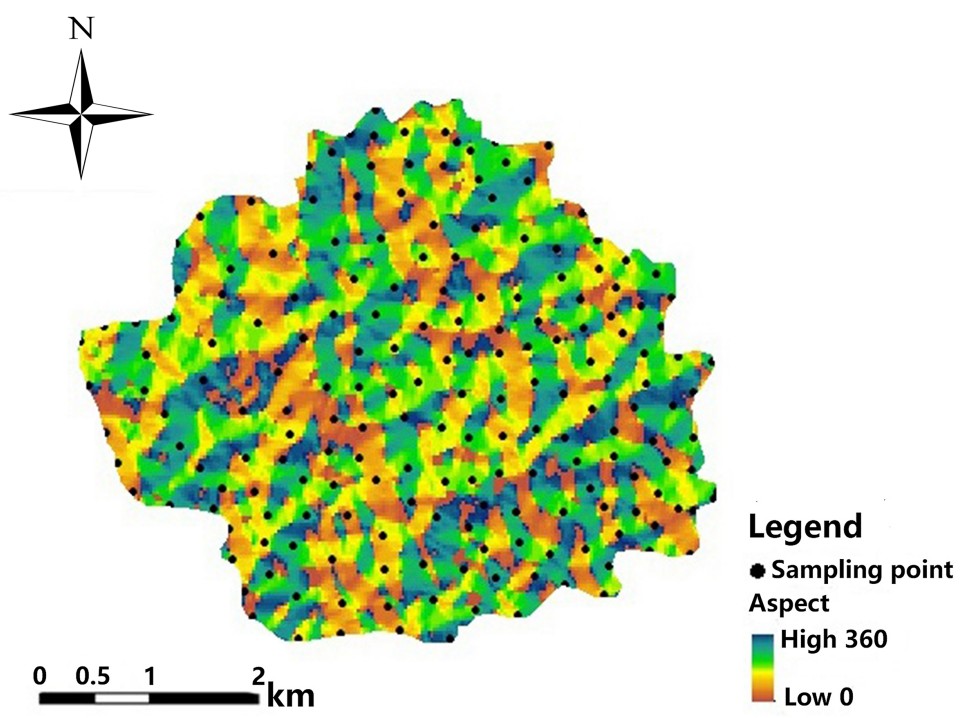

**Figure 10** Slope distribution map.

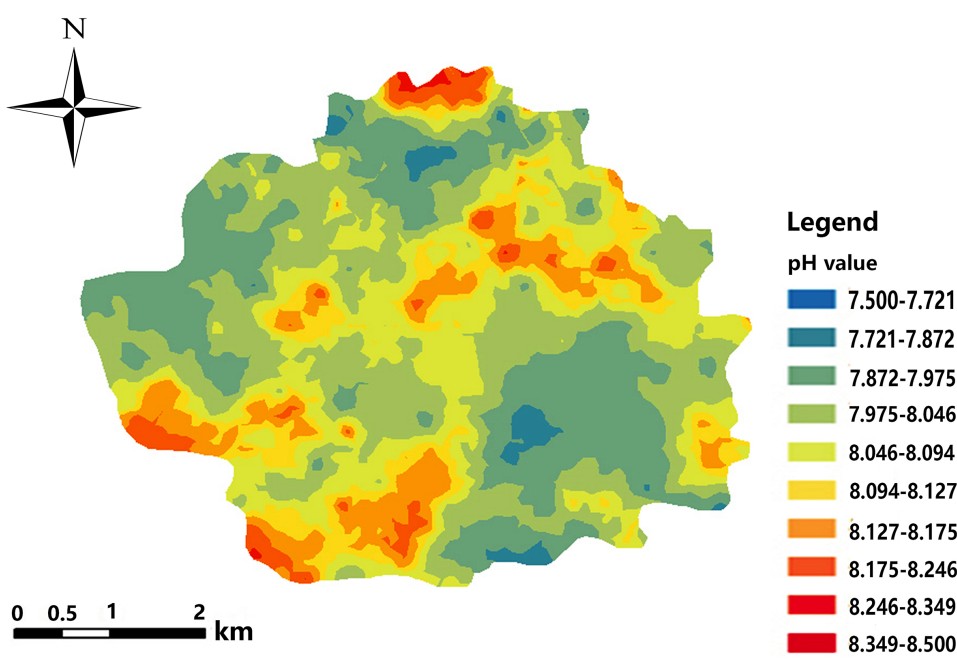

**Figure 11** Spatial distribution map of soil pH.

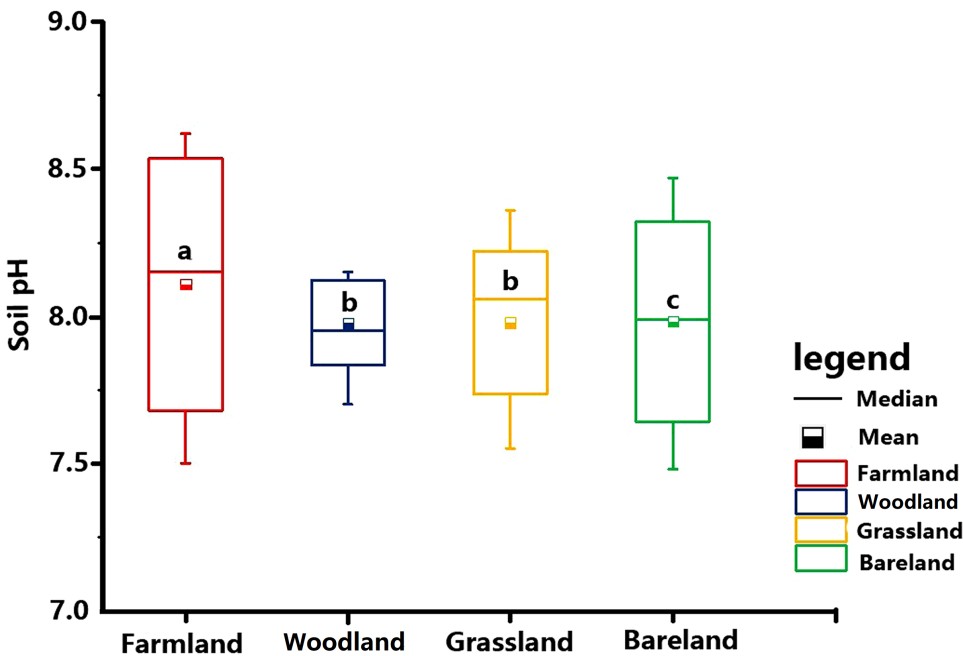

**Figure 12  Different land use patterns of soil pH.**

**Table 5  Semivariogram models and model parameters for soil properties in four land uses.**

| Land use patterns | Theoretical model | Coefficient of variation (%) | Nugget ($C_0$) | Sill ($C_0+C$) | Nugget/Sill (%) | Range (m) | Determining coefficient ($R^2$) |
|---|---|---|---|---|---|---|---|
| Farmland | Spherical model | 17.25 | 0.22 | 0.37 | 59.15 | 195 | 0.62 |
| Forestland | Spherical model | 17.09 | 0.31 | 0.48 | 63.49 | 180 | 0.58 |
| Grassland | Spherical model | 16.95 | 0.21 | 0.34 | 62.12 | 175 | 0.56 |
| Bareland | Spherical model | 14.21 | 0.19 | 0.29 | 65.59 | 181 | 0.59 |

use patterns varied from 8.09 for farmland to 7.98 for bare land, 7.97 for grassland and 7.96 for woodland. A comparison of the pH values in farmland and woodland topsoils shows that the pH value of woodland is lowest. An explanation for this might be that the tree species on woodland is Masson pine (Pinus massoniana Lamb), which has an acidifying effect on soil.

At 14.21%, the CV of pH is lowest on bare land; the pH of grassland and woodland is lower than that of farmland (Table 5). However, previous research established that the pH of woodland and farmland had the lowest CV, which could be the result of the uniform conditions in the region, such as small changes in slope and its direction, leading to a uniformity of soil (*Cambardella et al., 1994*; *Kavianpoor et al., 2012*; *Jeloudar et al., 2014*). The possible reasons require further investigation.

On the whole, the spatial distribution of soil pH is closely related to land use (*Mao et al., 2014*). This might be caused by the application of urea fertilizer, which has been proven to increase the soil pH (*Petrie & Jackson, 1984*).

Numerous studies have shown a decreasing soil pH with increasing number of cropping years (*Meng, Li & Liu, 2000*; *Zhao, Wu & Liu, 2000*). The average soil pH is the highest in farmland, followed by grassland and bare land, and the average pH in woodland is lowest. By studying the spatial variability of soil properties in an Alfisol soil catena, *Rosemary et al. (2017)* arrived at similar conclusions, namely that soil pH in paddies is high.

Influence of aspect on the spatial distribution of soil pH needs further research.

## CONCLUSION

The investigated parameters follow a normal distribution. For pH, the best-fitting variogram model was a spherical one. A practical application of our research results may be that the cost of the production cycle can be reduced by the inclusion of the models we established for application in directional semivariograms in interpolation analysis, improving the reliability of local assessments of the analysed soil pH. In order to reduce production costs, a sampling interval of 80–100 m is recommended for soil pH. The spatial distribution maps based on the kriging interpolation method were successfully applied in soil pH studies.

In this study, the CV is 17.18%, which can be classified as moderate variation, and is the result of both structural factors (parent material, topography, climate) and random factors (soil biology, human disturbance, sampling design and measurement error). This study focused on the spatial variability of soil pH as a result of the interaction of topographic factors, soil and land use patterns. In general, studying the spatial variability of soil pH can provide a theoretical basis for the restoration and improvement of soil quality, including the rapid restoration of soil in red-bed ecosystems and ecological reconstruction in the moist environment of south China.

### Funding

This work was supported by the National Natural Science Foundation of China (NO. 41771088 and NO. 51779279) and the Special Project for Key Basic Research of the Chinese Ministry of Science and Technology (NO. 2013FY111900). The funders had no role in study design, data collection and analysis, decision to publish, or preparation of the manuscript.

### Grant Disclosures

The following grant information was disclosed by the authors:
National Natural Science Foundation of China: 41771088, 51779279.
Key Basic Research of the Chinese Ministry of Science and Technology: 2013FY111900.

### Competing Interests

The authors declare there are no competing interests.

## Author Contributions

- Ping Yan conceived and designed the experiments, performed the experiments, analyzed the data, contributed reagents/materials/analysis tools, prepared figures and/or tables, authored or reviewed drafts of the paper, approved the final draft.
- Hua Peng conceived and designed the experiments, performed the experiments, analyzed the data, contributed reagents/materials/analysis tools, prepared figures and/or tables, authored or reviewed drafts of the paper, professor Peng died, unfortunately, but he made the same contribution as the corresponding author.
- Luobin Yan conceived and designed the experiments, analyzed the data, contributed reagents/materials/analysis tools, prepared figures and/or tables, authored or reviewed drafts of the paper, approved the final draft.
- Shaoyun Zhang performed the experiments, analyzed the data, contributed reagents/materials/analysis tools, approved the final draft.
- Aimin Chen performed the experiments, analyzed the data, contributed reagents/materials/analysis tools, prepared figures and/or tables, approved the final draft.
- Kairong Lin authored or reviewed drafts of the paper, approved the final draft.

## Field Study Permissions

The following information was supplied relating to field study approvals (i.e., approving body and any reference numbers):

The study area in Nanxiong Basin does not belong to an institute or agency, thus, a field permit was not necessary.

## Data Availability

The raw measurements are available in Supplemental Information 1.

## Supplemental Information

Supplemental information for this article can be found online at http://dx.doi.org/10.7717/peerj.6342#supplemental-information.

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
