# Peer review of "Spatial variability in soil pH and land use as the main influential factor in the red beds of the Nanxiong Basin, China"

_PeerJ, doi:10.7717/peerj.6342_

## Round 0.1 · original submission · Major Revisions

Both reviewers suggest that the manuscript touches an interesting topic, but have expressed significant concerns as well. Please use the reviewers' comments as guidance to revise your manuscript. Once the revision is done, we will re-review the manuscript and judge from the reviewers' comments to see where it goes.

·

Basic reporting

Your points are clear enough but there are significant language usage problems throughout this manuscript. The following are some examples and my reasons for pointing them out: "Element imbalances" (line 34) is too vague to convey that soil pH is important to plants. "Inhomogeneity" (line 35) should be avoided in this case because it is inappropriate to use an antonym negated with a prefix of "un-" or "in-" to define a term, and also some complex systems can be homogeneous. This and other paragraphs of the introduction clearly are tackling the problem of establishing causation between soil pH and vegetation, but the contrived language tends to mask the valid points you have raised.

You could avoid colloquialisms like "hot topics" (line 42) or "so far" (line 75) in scientific writing.

Many of the paragraphs appear truncated, leaving me wondering why the claims you make are true and relevant. A good example: Why did "scholars worldwide" (line 44) being applying geostatistics? This is a great place to make a concise lists of the pros (and minor cons) of using this suite of statistical tools.

I've often seen geostatistics incorporating or correlating two or more properties, which contradicts your claim (line 52) that geospatial properties "are rarely analysed in combination".

I'm confused by the phrase "soil parameters are highly variable in space and time" (line 55) because soil properties tend to change very slowly. Short-range variability of soil pH is very high but not over time (on the human scale).

I understand your points, which are appropriate, but adverbs customarily come before the word they qualify. The sentence from lines 58 to 60 maybe should read, "Because the spatial distribution of soil pH is stochastic, accurately measuring the spatial distribution of soil pH has implications for crop production." This avoids convoluted and/or redundant wording to convey your point.

It is atypical for any scientist to "prove" a hypothesis (line 60) but rather suggest a hypothesis that has not been disproved.

"Structure" (lines 59, line 65, etc.) has a number of very different meanings in statistics and soil science. It is good that you have suggested this as a factor, but you could define this term so readers of different backgrounds can better understand you. Likewise, the term "rational" can be used in many different ways, and you could include a definition or exclude the term as it can be assumed your readers are rational.

"Redbed" (line 66, etc.) requires definition, additionally "purple soil". These are not terms that are internationally used.

The term "eco-security" is somewhat unusual and vague. It is unclear whether you mean the security of the ecosystem or security of the soil, which you mention erodes. This is, of course, another very valid point to raise, but your wording makes it unclear what exactly you wish to convey regarding the security of an ecosystem.

"Past studies have demonstrated that the extent of soil erosion by water varies with pH" (line 72) is a very important point, but a description of the mechanism by which this occurs would be very beneficial to your argument for the importance of soil pH to soil erosion.

I appreciate the organization of your objectives (line 79). In the first objective (i), you say "status" when I think you mean "value", because the pH meter yields a value and not a status. Objectives ii and iii appear to be so similar that they could be combined into one objective. It may make more sense to state your first objective and then a hypothesis of what properties are influencing the values you find.

Experimental design

I'm familiar with the "nested sampling method", and I can imagine many ways it is appropriate for this study. However, you could state this explicitly in your methods.

There is insufficient detail in the soil sample collection section of the "Research method" section to reproduce the fieldwork and labwork. Which crops were harvested? Which "soil types" determined your sampling densities? Was the water used tap water or DI water? You have listed the specifications of the pH meter but not the specifications of the pH glass electrode, if that's what you've used. What other properties of the soil did you measure? Your objectives state you wish to determine which properties determine soil pH, but only soil pH was measured. Did you measure topographical data such as slope angle and aspect? These are required for your conclusions (line 283).

Validity of the findings

Your production of the semivariogram is very good and appropriate for assessing spatial variability of soil pH values.

Sections of your results entitled "Analysis of influential factors", "Topographic factors", and "Land use pattern" are not results of your study and belong in your discussion. Discussing potential factors does not add data to your study, and to infer that these are influential to soil pH in this study without collecting data regarding these variables is inappropriate.

I'm unsure the following result is supported by a data set absent slope data of each value of soil pH: "The pH of soil is highest on the downslope, followed by the middle slope, and is lowest on the upper slope." (line 258)

Your conclusions fit the results that are based specifically on your data, but the statement about "reducing the cost of the production cycle" (line 279) is unclear. Production cycle of semivariograms?

It is clear that "soil pH is affected by both structural and random factors" (line 283), but this conclusion can be drawn from a wide range of coefficients of variation, as listed in the assumption in this model from your methods.

Additional comments

Thank you for your work and effort to characterize soil pH in such a large and interesting region of China. Your work is important to the expansion of our knowledge of these soils, whose properties are worth studying in greater depth. I have included in my review of your manuscript a critique of your basic reporting, but it is not intended to detract from your message and review of the literature, both of which are valuable to the manuscript overall but somewhat lost in somewhat convoluted language I found hard to understand. The organization and causal inferences of this manuscript are its greatest weaknesses, but its novelty and value to the mapping of these soils are its greatest strengths.

·

Basic reporting

This manuscript is aimed to investigate the Spatial variability of soil pH and land use as the main influential factor in redbeds of the Nanxiong Basin, China.
The writing is clear enough, however, the manuscript contains a number of spelling and grammatical errors which should be carefully revised before acceptance. The literature is suitable for the purpose of the article.

Experimental design

no

Validity of the findings

no

Additional comments

Overall, the manuscript is publishable, but, needs a critical major revision first. The minor comments are mentioned in the attached PDF and the major comments are outlined below:

1. The section of discussion is not clear. Please add the contents of comparative analysis and the disadvantage of this study in the manuscript:
2. The authors explained too much in some parts of the manuscript. However, the influential factor is not clear enough. These sections should be rewritten
3. The materials and methods should be explained properly in the Research method section.

---

## Round 0.2 · Minor Revisions

Please follow the reviewer's suggestions to include a map of the area.

·

Basic reporting

Major revisions I suggested in my first review have been satisfactorily addressed.

Experimental design

Major revisions I suggested in my first review have been satisfactorily addressed.

Validity of the findings

Major revisions I suggested in my first review have been satisfactorily addressed.

Additional comments

Thank you for addressing my first review's comments.

·

Basic reporting

no

Experimental design

no

Validity of the findings

no

Additional comments

Most comments are well responsed.
But the authors mentioned the land use and land cover for many times, but there is no related support information. A map of land use/ cover classification in the study area would be helpful to understand the text.
This comment should be addressed properly as this is the base of the comparative analysis in this study.

---

## Round 0.3 · accepted · Accept

The revised manuscript has successfully addressed the reviewers' comments. I now believe it is ready for publication.